# Demographic and Health Indicators in Correlation to Interstate Variability of Incidence, Confirmation, Hospitalization, and Lethality in Mexico: Preliminary Analysis from Imported and Community Acquired Cases during COVID-19 Outbreak

**DOI:** 10.3390/ijerph17124281

**Published:** 2020-06-15

**Authors:** Nina Mendez-Dominguez, Alberto Alvarez-Baeza, Genny Carrillo

**Affiliations:** 1Department of Health Sciences, School of Medicine, Universidad Marista, Periférico Norte Tablaje Catastral 13941, Merida 97300, Mexico; nmendez@marista.edu.mx (N.M.-D.); abaeza1094@gmail.com (A.A.-B.); 2Department of Environmental and Occupational Health, School of Public Health, Texas A&M University, 212 Adriance Lab Road, College Station, TX 77843, USA

**Keywords:** sentinel surveillance, COVID-19, Mexico, disease outbreak, epidemics

## Abstract

This study’s objective is to analyze the incidence, lethality, hospitalization, and confirmation of COVID-19 cases in Mexico. Sentinel surveillance for COVID-19 cases in Mexico began after the confirmation of the first patient with community transmission. Methods: This epidemiologic, cross-sectional study includes all clinically suspected, and laboratory-confirmed cases nationwide from the beginning of the outbreak to 21 April 2020. State-cluster demographic data and health indicators were analyzed in reference to epidemiologic measures, with logistic regressions for the dependent variables of incidence, confirmation, and lethality. Results: The national incidence was 13.89/100,000 inhabitants with a 6.52% overall lethality and a confirmed-case mortality of 11.1%. The incidence variation significantly correlated with migration, but not urbanization. Pediatric patients were less prone to be tested (OR = −3.92), while geriatric individuals were a priority. State lethality positively correlated with the proportion of the population assisted at public hospitals and correlated inversely to the number of hospitals and clinics in the state. Conclusions: Migration strongly correlated with incidence; elderly patients had lower odds of being hospitalized but were likely to die. Patients aged <15 were less prone to be laboratory-confirmed. Case confirmation was not performed in all hospitalized patients, but 72.15% of hospitalized patients had favorable outcomes to date.

## 1. Introduction

In December 2019, a unique disease (COVID-19) caused by a novel coronavirus, SARS-CoV-2, appeared in Wuhan, Hubei Province, China [1]. It has spread to other areas of China and more than 185 countries and regions around the world [2,3,4]. On 11 February 2020, the World Health Organization (WHO) named the disease caused by 2019-nCoV: Coronavirus Disease-2019 (COVID-19) [4]. COVID-19 has an average incubation period between 3 and 7 days, with one day as the shortest and 14 days the longest. On 30 January 2020, the WHO declared the COVID-19 outbreak as the sixth public health emergency of international concern, following H1N1 in 2009, polio in 2014, Ebola in West Africa in 2014, Zika in 2016, and Ebola in the Democratic Republic of Congo in 2019. Health workers, governments, and the public needed to cooperate globally to prevent its spread. In Latin America, the first case was reported in Brazil on 25 February 2020, and in Mexico, on the 28th, three days later [5]. With the introduction of the first case, Mexico entered the imported case transmission stage of the epidemic. On 24 March, the confirmation of domestic transmission and numerous suspected cases marked the onset of domestic, community transmission. With community transmission, the health authorities implemented mandatory sentinel surveillance. COVID-19 strategic objectives set by WHO include interrupting human-to-human transmission (including reducing secondary infections among close contacts and health care workers), preventing community transmission, and preventing further international spread [6]. Thus, migration, population density, access to health services, availability of medical resources, and infrastructure are relevant demographic characteristics that may define the course of COVID-19 in Mexico, as in many other countries [7].

Mexico experiences significant disparities in demographic and social aspects per sub-region. Mexico comprises 32 states with different ethnic, social, and economic characteristics that may translate into a differential epidemiologic profile of COVID-19, in both incidence and lethality [8].

Sentinel surveillance is employed in Mexico for epidemiologic purposes, as has been used before for dengue, chikungunya, and influenza AH1N1 [9,10]. For COVID-19, sentinel surveillance represents a significant cost reduction and is useful for estimating the affected population, but on the other hand, it implies that many patients will not be sampled and may not have a precise diagnosis. On epidemiologic grounds, it is yet unknown how accurately sentinel surveillance (as a subsample) may represent the universe of infected individuals. Therefore, the objective of the present study is to analyze the incidence, lethality, hospitalization and confirmation of COVID-19 cases in Mexico from the beginning of the outbreak to 21 April, clustering analysis by state concerning healthcare access, availability of medical resources, and demographic indicators. We also analyzed case by case the confirmation, hospitalization, and death association to patients’ age, gender, and migratory status.

## 2. Materials and Methods

### 2.1. Study Setting and Data Sources

This observational, epidemiologic cross-sectional study includes all cases of COVID-19 reported by the Mexican General Council of Epidemiology (DGE). Such data comprise probable and confirmed cases since the beginning of the outbreak to 21 April 2020, after sentinel surveillance was established. 

A suspected case pertains to a patient with an acute respiratory illness that has been in contact with a confirmed or probable COVID-19 case in the last 14 days before symptom onset. It also includes a patient with severe acute respiratory illness (fever and at least one sign/symptom of respiratory disease, e.g., cough, shortness of breath, requiring hospitalization) in the absence of an alternative diagnosis that thoroughly explains the clinical presentation. In Mexico, suspected cases are those with symptomatology and clinical manifestations consistent with COVID-19, that were not laboratory-tested. A confirmed case is one that tested positive for COVID-19 in a laboratory test, and those tested negative were excluded from the present study. In Mexico, RT PCR, which is a real-time reverse transcription polymerase chain reaction (rRT-PCR) test for the qualitative detection of nucleic acid from SARS-CoV-2 in upper and lower respiratory specimens is the approved test for diagnosis nationwide to date.

Data were obtained from open-access epidemiologic data of COVID-19 available from the DGE webpage, including laboratory-confirmed and clinically diagnosed (suspected) patients. Sociodemographic characteristics of patients such as age, gender, state of residence, and confirmation status were obtained from all reported cases. Descriptive statistics data were obtained by state-clusters and include frequencies, percentages, indexes, central tendency, and dispersion measures. Mexico has 32 states and 1 capital city; clustering reported by each state made it possible to analyze demographic and population health indicators using the state-clusters as study units. 

Case by case data were obtained from the open-access DGE dataset for COVID-19 during the imported case and community transmission phases, including all registries up to 21 April 2020. In our analyses, variables regarding age, gender, status at discharge, migratory movements, laboratory confirmation, hospitalization, and outcome were included.

### 2.2. Measurement

The demographic and population health services indicators that pertain to each state of Mexico were selected for the present study:(a)Population size was estimated from the official data from the National Population Council (CONAPO) for 2020 based on the latest available census developed by the National Institute for Statistics and Informatics (INEGI). Population size was used as a crude measure for calculating incidence but transformed into percentiles as a proxy for population size. Population average age, population proportion by gender, and population proportion of pediatric (≤15 years) and elderly (≥65 years) were based on CONAPO´s estimates [11]. Population size was also included in percentiles for state-cluster description.(b)The indigenous population is an essential demographic indicator in the study of health determinants in Mexico, and it was obtained from INEGI representing the proportion of the population belonging to an indigenous ethnicity [12].(c)Urbanization level. This quantitative variable was obtained from INEGI state data on the percentage of the population living in communities with urban services and infrastructure.(d)Migration. This includes the totality of national and international migratory movements projected for the year 2020 from CONAPO estimates [11].(e)The percentage of the population without access to private medical care, only public care, was obtained from the General Board of Health Information (DGIS).(f)The number of health facilities, including public and private clinics and hospitals per geographic unit, was obtained from DGIS [13].(g)Confirmed and suspected cases based on sentinel surveillance. A subsample of patients would be laboratory-confirmed instead of testing every patient that meets clinical criteria for COVID-19. All untested patients are considered clinically diagnosed and labeled as suspected cases [14].(h)The state incidences were obtained by multiplying the number of cases (confirmed and suspected) by 100,000 and dividing by population size.(i)The state-cluster confirmations were obtained by dividing the number of confirmed cases in the state multiplied by 100, and then dividing it by the sum of both confirmed and suspected cases.(j)The state-cluster hospitalizations were obtained as a percentage of patients that were treated in the hospitals as a dichotomous variable, in contrast to ambulatory patients.(k)The lethality was calculated by obtaining the percentage of deaths divided by the sum of lab-confirmed and suspected cases.

In a case by case analysis, age was divided into three groups (<15, 15–65 and >65), gender was treated as a binary variable (female/male), and interstate migratory movement of the patient was considered as dichotomous, that is, indicating whether patients received medical attention in their residential state or not. 

The COVID-19 confirmation status was considered as a dichotomous variable (confirmed/suspected), and hospitalization (hospitalized/ambulatory) and survival outcome (death/recovery) variables were further considered as dependent variables for binary logistic regression models. 

## 3. Statistical Analysis

### Descriptive Statistics 

Population health indicators, incidence, confirmations, and lethality are presented in the following tables as frequency measures; median and interquartile range (25th and 75th percentiles) are shown in Table 1 along with W Shapiro tests for normal distribution, where *p* > 0.05 indicated normality. From case by case data, we present proportions, standard error, and confidence intervals (all shown as percentages) for age groups, gender, migratory status, confirmation, hospitalization, and survival outcomes. Our results are divided by confirmation, hospitalization, and survival outcome groups. Poisson regression models were used to assess state-cluster indicators and incidence, confirmation, or lethality as dependent variables, after adjusting by population age composition and gender, where the incidence rate ratio (IRR) was the measure of association (reference value of IRR = 1.00). In the case by case analysis, logistic regression for binary dependent variables was used to establish associations between age, gender, and place of residence of patients with the confirmation status, hospitalization, and fatal outcomes; odds ratio was the measure of association, where OR = 1.00 was the reference value. All statistical tests were developed using Stata 14^®^.

## 4. Results

The first patient with COVID-9 in Mexico was laboratory-confirmed on 28 February, corresponding to a patient who arrived from Italy on 21 February, and showed symptoms that same day. Between 28 February and 16 March 105 patients reported symptom onset. They were confirmed as imported cases: more than 50% (*n* = 53) corresponded with travelers arriving from Spain, 28.6% (*n* = 30) from the U.S., 9.5% (*n* = 10) from Italy, 5.7% (*n* = 6) from Germany, 3.8% (*n* = 4) from France, with one case from Singapore and one from Cuba. 

On March 24, health authorities in Mexico declared COVID-19 community transmission, and sentinel surveillance was implemented, meaning that a subset of patients would be laboratory-tested instead of all patients with compatible clinical manifestations. National incidence was 13.89 per 100,000 inhabitants; overall case lethality was 6.52% and confirmed-case lethality was 11.1%, showing with variations between the Mexican states. Incidence per state-cluster is shown in Figure A1. COVID-19 cases in the studied period were 55.76% female, and the mean age was 43.4 years (±0.49). An amount of 2.8% (4.97) of patients were aged <15, and 12.2% (2159) were aged 65 and over; by 21 April, 17,763 cases had been registered in the 32 states.

Age distribution did not significantly differ between states or by gender (male 43.8 years old, female 44.8 years old). Description of state-cluster indicators and COVID-19 cases per state-cluster are presented in Table 1.

In Table 1, median values of hospitalization surpass the median of confirmations, suggesting the possibility that a proportion of hospitalized patients remain untested. Median of state-cluster incidences per 100,000 inhabitants was 7.62, while overall COVID-19 mortality was 0.80 per 100,000 inhabitants in the studied period. Confirmed COVID-19 patients summed 9501, and 8262 patients were untested remaining as suspected. 

In the Poisson regression model (Table 2), a significantly higher incidence rate ratio was observed with a higher migration rate (IRR = 6.43:1), but urbanization was minimally (yet significantly) associated with IRR. The availability of clinics and hospitals, indigenous ethnicity, and population affiliated to public health institutions, correlated inversely with confirmation and lethality in state-clusters, while urbanization correlated with a more significant confirmation in both. Hospitalizations varied among state-clusters, migration and availability of clinics and hospitals were inversely associated, and urbanization and population affiliated to public health institutions showed higher IRR for hospitalization.

In the case by base descriptive statistics (Table 3), female patients were less frequently confirmed and treated ambulatory, but more deaths occurred among women. The average age of confirmed patients was younger than their confirmed peers, hospitalized patients were younger than ambulatory patients were, and fatal outcomes occurred among individuals aged 58.8 years on average. The proportion of patients who received medical attention while interstate migrants were 7% of all cases was averaged and represents 5.77% of confirmed cases, 6.5% of all hospitalizations, and only 4.75% of COVID-19 deaths. Confirmation was proportionally lower among hospitalized patients than ambulatory cases, and more than 88% of all fatal cases were laboratory-confirmed. Hospitalized cases were non-fatal in 72.15 cases, even when 37.24 remained as suspected.

Lethality differed by state-cluster, ranging between 0 and 25 percent, as shown in Figure A2. Lethality positively correlated with access to public health institutions and age, and inversely correlated to the female gender, availability of clinics/hospitals, and age. Nevertheless, the population ≥ 65 did not proportionally differ among patients with fatal outcomes. The correlation between demographic indicators of state-clusters and lethality are shown in Table 3.

In case by case, the logistic regression model (Table 4) shows that males were more prone to be hospitalized but also had fewer odds of being confirmed by the laboratory and with more fatal outcomes. In the cases of individuals aged <15, they were significantly less likely to be laboratory-confirmed, but also had fewer odds for hospitalization or death. Patients aged >65 were less likely to be hospitalized but have significantly higher odds of a fatal outcome. Interstate migrants were less prone to be laboratory-confirmed but showed more susceptibility to hospitalization and death. Patients with fatal outcomes were those with more odds of being laboratory-confirmed.

## 5. Discussion

We have presented a preliminary overview of COVID-19 incidence, confirmation, hospitalization, and lethality distribution by state-clusters in Mexico during the pandemic, in association with state-clusters demographics and health indicators. Previously, Gomez Dantés and colleagues reported that Mexico and its 32 states are experiencing epidemiological transitions, highlighting relevant disparities between states [15]. These authors also stated how the convergence of communicable and non-communicable diseases was expected to affect the response capacity and performance of the health system if trends remained stable. In the present study, we observed that confirmation and hospitalizations were both more frequent in the states where more clinics and hospitals are available. At the same time, a proportion of patients moved from their state of residence to seek medical attention, according to the case-by-case results. Although they might seem discordant, our results show a lower confirmation and hospitalization in state-clusters with more availability of clinics and hospitals. Furthermore, this could be related to the fact that the national strategy for the COVID-19 contingency explicitly asked that only public laboratories were allowed to perform diagnostic tests. In addition, it was established that the hospitals allowed to provide medical assistance to COVID-19 patients were mandatorily designated and, initially, those hospitals were only public institutions. Nevertheless, with the growing impact of the epidemic in Mexico, authorities later allowed some certified private laboratories and hospitals to perform the COVID-19 test [13].

Sentinel epidemiologic surveillance for COVID19 was planned to replicate the surveillance method initially implemented with influenza in 2009, involving the random subsampling of a proportion of all clinically cases, and laboratory testing for all hospitalized patients. Nevertheless, in the present study, in the case-by-case analysis, results suggest that not all hospitalized patients were laboratory-confirmed. The implications for not testing all hospitalized patients are diverse, providing a bias for statistical and administrative purposes. Furthermore, it affects the clinical approach and resources allocated to treat a patient that could have COVID-19 infection, as well as other patients showing symptoms associated with pneumonia, influenza or other viral or bacterial infections [16].

The incidence was higher among states with migration that is more significant in international borders and positively correlated with total migration. According to Ebrahim et al., the cancellation or suspension of mass gatherings and traveling would be critical to pandemic mitigation; according to our results, suspension of international traveling could have also been essential in the spread of the disease in Mexico [17]. 

In the present study, the urbanization did not correlate with higher incidence, confirmation, or total migration. Such results could be explained by the effectiveness of self-isolation strategies implemented before travel and migration were limited. The association between confirmation and access to public healthcare may reflect regulatory measures to limit the use of the only public Mexican laboratory for COVID-19 tests. Nevertheless, states with higher population access to public healthcare had greater lethality than those with access to private healthcare. The lethality may reflect the inequities present in public healthcare, such as a lack of resources to ventilators and medication, as well as higher number of patients seeking public healthcare, possibly prompting increased demand for more robust public services. These findings are the opposite to what Verdery and Smith-Greenaway in neighboring Guatemala published, where confirmed cases were more common in the most densely populated administrative area [18]. In our results, confirmations were more frequent in the less populated areas due to the sentinel surveillance implementation. 

The younger population (≤15 years) has a significantly lower confirmation, as they are three times less likely to be laboratory-tested. These findings may be related to the expected increased rate of complications, severity, and mortality among patients aged 60 and over, but also may generate bias if fatalities occur, as standardized age lethality is based only on confirmed cases [19,20]. If this problem remains, severity and mortality among young patients may continue underestimated due to sampling methods. Among patients aged 65 and over, our analysis suggests a higher confirmation, but less hospitalization and higher odds for death. Hospitalized patients were more likely to non-fatal outcomes, but older patients had fewer odds of hospitalization. Older individuals are more prone to have more comorbidities than their younger peers; however, Mexico has a younger age of onsets of non-communicable diseases, such as obesity, diabetes, hypertension, and chronic kidney disease. Therefore, the prevalence of comorbidities may be a determinant of severity among older adults and may reflect an increasing need for in-hospital medical attention [15,21]. 

Rodriguez-Morales et al. suggest that the impact of the COVID-19 pandemic would be less devastating in Mexico than in other Latin American countries. However, lethality in Mexico seems to be higher in the states with more population affiliated to public healthcare access and a limited private medical infrastructure. Recommendations from WHO for those patients with symptoms to avoid more hospitalizations are to stay at home in self-isolation and contact their healthcare provider by phone, instead of going directly to clinics to avoid overuse and saturation of public medical infrastructures. Therefore, this measure may prioritize and improve health services for severe cases, especially the most vulnerable patients living in resource-limited areas of Mexico [6,22]. 

The results we present are based on the first weeks of the epidemic in Mexico, which did not correspond chronologically to other regions of the world. Therefore, incidence comparisons in the studied period between Mexico and other countries may not be appropriate. Lethality in Mexico for the present communication was estimated using both suspected and confirmed cases; thus, it may seem incomparable to the standard estimations worldwide, where confirmed-case mortality is commonly employed. The observed confirmed-case mortality of 11.1% in Mexico may mean that Mexico ranks fourth place among the world’s countries with the highest mortality, only after France (25.4%), Italy (14.4%) and United Kingdom (14.2%), and equal to Sweden—all European countries where the COVID-19 epidemic started earlier. In consequence, Mexico is the country with the highest confirmed-case mortality in Latin America, followed by Ecuador (8.5%) and Brazil (5.6%). Nevertheless, these comparisons should be carefully considered as it has been described how in places where testing rates are lower, case-confirmation lethality may be overestimated [23]. Inaccuracy occurs when the real number of cases remains unknown or miscalculated [24] and because elderly patients with severe manifestations are more prone to be tested. In sentinel surveillance, the most vulnerable patients are prioritized for sampling due to their conditions by definition. Quite the opposite is the case of Germany, where a lower case-fatality has been reported as a result of implementing a broad testing strategy that includes sampling also non-severe cases and younger people [25]. By March 2020, the WHO director stated that around 3.4% of registered COVID-19 cases globally died [26]. However, before such a statement, an approximation of 2% was estimated, meaning that lethality rates may still be changing on account that patients may not have recovered entirely or may even be hospitalized [27]. As with other emerging diseases, global lethality rates of COVID-19 will continue to be uncertain due to a number of factors, including the timing of the outbreak in the different regions of the world, the severity of clinical manifestations between human groups and even the possibility and timing of virus mutations [28]. A clinical series study conducted in Wuhan included patients with and without comorbidities, and a median age of 56 and a lethality of 4.3% was observed [29]. Therefore, it could be considered that overall lethality in Mexico is, to the studied date, higher than that projected with the WHO estimates, and higher than that observed so far in other Latin American countries.

### Limitations

The present study, as any other cross-sectional design, has particular limitations: firstly, those that derive from the timing of the COVID-19-related outcomes. The demographic migration indicators also have limitations, as they pertain to projection and estimations but may exclude illegal migratory movements. The health infrastructure, such as clinics and hospitals, are indicators of availability of health resources, but may not reflect the actual access to those resources for diagnosis, as only a few laboratories nationwide have been approved for testing for COVID-19. The incidence and confirmation, as in other similar COVID-19 studies, may underestimate the true incidence and confirmation due to underreporting and asymptomatic carriers. Lethality may be considered with caution, as it may vary from official registries, as authors employed both confirmed and suspected cases as the denominator. Finally, even after adjusting cluster-state analysis by population age and gender composition, we did not adjust our results according to comorbidity distribution in the studied population. 

## 6. Conclusions

From the beginning of the outbreak, migration was significantly associated with incidence ratios according to state-cluster analyses. Confirmation was not performed in all hospitalized patients, but 72.15% of hospitalized patients had positive outcomes to date. Elderly patients had lower odds of being hospitalized, but were likely to die, while interstate migrants had more propensity to fatal outcomes, yet were less likely to be laboratory-confirmed.

Age group laboratory-testing, if not corrected, could result in biased assumptions of severity and lethality among young patients. These findings may help health professionals and policymakers to consider the importance of maintaining the shelter-in-place policy to prevent further exposure to communities not currently affected. It is imperative to continue emphasizing nationwide preventative measures to avoid a further increase in infection and lethality. Additional studies need to be completed in future months to learn more about COVID-19 symptoms and clinical manifestations in Mexico and the population most affected. 

## Figures and Tables

**Table 1 ijerph-17-04281-t001:** Descriptive statistics of Coronavirus (COVID-19) state-clusters (*n* = 32 states) in Mexico between 28 February and 21 April 2020.

State-Cluster Indicators	Median	Interquartile Range	25th Percentile	75th Percentile	Shapiro–Wilk W Test
Population size	3,121,544	3,421,650	1,796,128	5,217,777	0.967
Population size in percentiles	18	20	10	30	0.466
Urbanization	39.43	21.78	33.45	55.22	0.946
Population affiliated to public Health institutions	48.08	25.035	38.305	63.34	0.959
Indigenous ethnicity	19.80	21.46	10.13	31.59	0.934
Number of clinics and hospitals (units)	50	78	30	108	0.966
Total migration rate 2020	75.26	0.62	74.89	75.51	0.427
Coronavirus (COVID-19) State-cluster epidemiologic measures					
Total cases	278	423	131	554	0.967
Incidence per 100,000 habitants	7.62	12.50	5.11	17.61	0.887
Confirmed cases	142	170	78	247	0.967
Confirmed cases aged < 15	2	3	1	4	0.386
Confirmed cases aged > 65	20	20	12	32	0.965
Hospitalization	188	275	78	353	0.967
Deaths (count)	14	30	6	36	0.962
Lethality	5.36	4.79	4.22	9.0	0.558

**Table 2 ijerph-17-04281-t002:** COVID-19 incidence, confirmation, hospitalizations, and lethality by state-cluster population indicators in Mexico between 28 February and 21 April (*n* = 17,763 in 32 state-clusters).

State-Cluster Indicator	Incidence Rate Ratio	Standard Error	Z	*p* Value	Confidence Intervals 95%	
Incidence (Number of cases)						
Migration	6.43	1.24	9.64	0.000	4.41	9.39
Clinics and hospitals	0.99	0.18	−6.74	0.077	0.92	1.01
Indigenous ethnicity	1.03	0.20	5.53	0.070	0.97	1.04
Population affiliated to public health institutions	1.01	0.11	1.91	0.056	0.98	1.02
Urbanization	0.98	0.00	−3.66	0.002	0.97	0.99
Confirmation						
Migration	1.05	0.00	26.39	0.054	1.05	1.06
Clinics and hospitals	0.79	0.00	−29.37	0.000	0.79	0.79
Indigenous Ethnicity	0.89	0.00	−31.30	0.000	0.88	0.90
Population affiliated to public health institutions	0.81	0.00	−70.79	0.001	0.79	0.81
Urbanization	1.63	0.01	137.28	0.000	1.62	1.64
Hospitalization (Hospitalized cases)						
Migration	0.65	0.04	−6.43	0.000	0.58	0.74
Clinics and hospitals	0.83	0.00	−26.52	0.000	0.83	0.83
Indigenous ethnicity	1.01	0.00	1.93	0.053	.99	1.02
Population affiliated to public health institutions	1.27	0.01	29.46	0.000	1.25	1.29
Urbanization	1.10	0.00	30.70	0.000	1.09	1.10
Lethality (Fatal cases)						
Migration	1.02	0.00	48.41	0.065	0.98	1.02
Clinics and hospitals	0.89	0.00	−57.32	0.000	0.79	0.79
Indigenous ethnicity	0.82	0.10	−13.69	0.090	0.88	1.01
Population affiliated to public health institutions	0.79	0.01	−35.99	0.000	0.78	0.80
Urbanization	1.58	0.02	42.07	0.000	1.55	1.62

Poisson regression by state-cluster population size and adjusted by population age and gender composition.

**Table 3 ijerph-17-04281-t003:** COVID-19 cases by laboratory confirmation status, hospitalization, and mortality in Mexico between 28 February and 21 April 2020 (*n* = 17,763).

Variable	Percentage	Standard Deviation	Confidence Intervals 95%		Percentage	Standard Deviation	Confidence Intervals 95%	
COVID-19 Diagnosis status	Confirmed	*n* = 9501			Suspected	*n* = 8262		
Female	55.31	0.50	54.33	56.29	51.19	0.56	50.08	52.29
Average age (*mean*)	46.43	0.55	45.35	47.50	42.34	0.51	41.35	43.34
Age < 15	1.31	0.12	1.08	1.53	4.51	0.23	4.07	4.96
Age > 65	13.55	0.35	12.86	14.23	10.55	0.34	9.89	11.22
Interstate migrant	5.77	0.26	5.52	6.03	8.16	0.28	7.88	8.44
Hospitalization	62.76	0.50	61.79	63.73	75.72	0.47	74.80	76.64
Death	9.02	0.29	8.44	9.60	1.36	0.13	1.11	1.60
Hospitalizations /Ambulatory	Hospitalized	*n* = 5545			Ambulatory	*n* = 12,218		
Female	52.14	0.45	51.25	53.03	63.73	0.65	62.46	64.99
Average age (*mean*)	40.65	0.14	40.39	40.92	52.60	0.23	52.15	53.05
Age < 15	3.08	0.16	2.77	3.38	2.18	0.20	1.80	2.57
Age > 65	6.33	0.22	5.90	6.77	24.98	0.58	23.84	26.12
Interstate Migrant	6.49	0.22	6.05	6.93	8.28	0.37	7.55	9.00
Confirmation	48.80	0.45	47.91	49.69	63.82	0.65	62.55	65.08
Death	0.83	0.08	0.67	1.00	15.64	0.49	14.68	16.59
Fatal outcome	Fatal	*n* = 968			Non-fatal	*n* = 17,763		
Female	68.32	1.50	65.39	71.25	55.03	0.38	54.28	55.78
Average age (*mean*)	58.88	0.47	57.96	59.80	43.54	0.13	43.30	43.79
Age < 15	0.52	0.23	0.06	0.97	2.93	0.13	2.67	3.18
Age > 65	38.08	1.56	35.02	41.14	10.66	0.24	10.19	11.13
Interstate migrant	4.75	0.68	4.06	5.43	7.18	0.20	6.98	7.38
Confirmation	88.44	1.03	86.43	90.46	51.47	0.39	50.71	52.23
Hospitalized	10.53	0.99	8.59	12.46	72.15	0.35	71.47	72.83

Proportions and dispersion measures are presented multiplied by 100.

**Table 4 ijerph-17-04281-t004:** Sociodemographic characteristics associated to COVID-19 confirmation, hospitalizations, and death in Mexico between 28 February and 21 April (*n* = 17,763 cases).

Outcome	Odds Ratio	Standard Error	Z	*p* Value	Confidence Intervals 95%	
Confirmation						
Male gender	0.92	0.03	−2.85	0.004	0.86	0.97
Age < 15	0.30	0.03	−11.48	0.000	0.24	0.36
Age > 65	0.92	0.05	−1.64	0.101	0.83	1.02
Interstate migrant	0.69	0.04	−6.16	0.000	0.61	0.77
Death	5.59	0.58	16.47	0.000	4.55	6.86
Hospitalization	0.65	0.02	−11.77	0.000	0.61	0.70
Hospitalization						
Male gender	1.54	0.06	12.07	0.000	1.44	1.66
Age < 15	0.91	0.10	−0.88	0.380	0.73	1.12
Age > 65	0.23	0.01	−28.33	0.000	0.21	0.26
Interstate migrant	1.36	0.09	4.68	0.000	1.19	1.54
Death	0.06	0.01	−25.17	0.000	0.05	0.08
Confirmed	0.65	0.02	−11.79	0.000	0.61	0.70
Fatal outcome						
Male gender	0.72	0.06	−4.26	0.000	0.62	0.84
Age < 15	0.55	0.25	−1.30	0.195	0.22	1.36
Age > 65	2.65	0.21	12.48	0.000	2.27	3.09
Interstate migrant	2.01	0.33	4.30	0.000	1.46	2.76
Confirmed	5.75	0.61	16.63	0.000	4.68	7.07
Hospitalized	0.06	0.01	−25.26	0.000	0.05	0.08

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
