# Peer review of "Demographic and Health Indicators in Correlation to Interstate Variability of Incidence, Confirmation, Hospitalization, and Lethality in Mexico: Preliminary Analysis from Imported and Community Acquired Cases during COVID-19 Outbreak"

_ijerph, 2020, doi:10.3390/ijerph17124281_

Round 1

Reviewer 1 Report

The authors try to analyse the incidence, lethality, and confirmation rate of COVID-19 cases in Mexico. However, the authors have not been successful in achieving their objective. In fact, I have comments, both, major and minor.

Major comments:

  • I am amazed at the authors’ use of linear regression. They should have used a Poisson regression with the number of cases, or deaths, as the dependent variable and population at risk as an offset, in addition to the explanatory variables. The most important consequence of not using Poisson regression is a specification error caused because the population at risk is not the same for all cases or deaths. It depends on, for instance, the sex and age structure of the area. This specification error leads to invalid inferences.
  • Authors should provide, perhaps in an appendix, residual analyses in order to check the absence of such specification errors.
  • Not all the demographic and population health services indicators measures are well explained. In particular, urbanization level is a qualitative or a quantitative variable? If it is quantitative, is measured as a percentage? Population without access to private medical care is a rate?
  • Authors wrote that carried out a descriptive analysis and also group mean comparison tests. However, I have not found any table with these analyses. Please include one table and explain in detail in the text.
  • All the variables have asymmetrical distributions and therefore mean and standard deviation are not very useful to summarise them. Authors should use median and quantiles instead.
  • Authors reported the number of patients in Phase 1, but not in Phase 2. Authors should provide this information because this has consequences on the inference. I mean if the number of patients is similar in both Phases, the statistical power of the analysis would be very limited.
  • Line 124: Authors wrote that they used logistic regression but thy not provide any evidence of it. I would like to see some table showing the results.
  • None of the figures cannot be interpreted. The possible patterns that can be observed actually respond to a larger population in these areas. Authors should represent the standardised rates by age and sex, at a minimum (maybe also by the migration structure).
  • Authors should explain in the limitations that they not control for all the confounders. They should mention the use of random effects in this case.

Minor comments:

  • Line 44: Reference 3 is missing.
  • Line 55: Authors wrote reference 6, but I cannot find refence 5 in the text.
  • Line 84: Authors should provide the web of the DGE.
  • Lines 84-85: Authors wrote that socioeconomic characteristics of patients and confirmation status was obtained from all reported cases. Authors should explain how they obtain this information. What are the sources?

Author Response

Dear reviewers, the authors want to thank you for your time, positive criticism, and insightfulness that has truly helped us improve our manuscript.

Please find changes in the revised version highlighted in yellow.

Reviewer 1

 The authors try to analyze the incidence, lethality, and confirmation rate of COVID-19 cases in Mexico. However, the authors have not been successful in achieving their objectives. In fact, I have comments, both, major and minor.

Major comments:

  • I am amazed at the authors’ use of linear regression. They should have used a Poisson regression with the number of cases, or deaths, as the dependent variable and population at risk as an offset, in addition to the explanatory variables. The most important consequence of not using Poisson regression is a specification error caused because the population at risk is not the same for all cases or deaths. It depends on, for instance, the sex and age structure of the area. This specification error leads to invalid inferences. Authors should provide, perhaps in an appendix, residual analyses in order to check the absence of such specification errors.

Thank you for your comment and your recommendation. We followed your suggestion of using Poisson regression with the population at risk as an offset; we understand the importance of avoiding specification errors and of preventing invalid inferences.

  • Not all the demographic and population health services indicators measures are well explained. In particular, urbanization level is a qualitative or a quantitative variable? If it is quantitative, is measured as a percentage? Population without access to private medical care is a rate?

Thank you, dear reviewer, we have corrected and completed information on all variables including urbanization and population affiliated to public health institutions.

  • Authors wrote that carried out a descriptive analysis and also group mean comparison tests. However, I have not found any table with these analyses. Please include one table and explain in detail in the text.

We have added descriptive statistics in tables, in response to your suggestion. Please find tables 1 and 3 in our results section.

  • All the variables have asymmetrical distributions and therefore mean and standard deviation are not very useful to summarise them. Authors should use median and quantiles instead.

Thank you, we have followed your recommendation, and we now present a table with median, interquartile range, and normality tests values.

  • Authors reported the number of patients in Phase 1, but not in Phase 2. Authors should provide this information because this has consequences on the inference. I mean if the number of patients is similar in both Phases, the statistical power of the analysis would be very limited.

Dear reviewer, to attend your comment accordingly, we expanded the study period to include data at the end of phase 2. Nevertheless, inattention to reviewer 2´s comments, we eliminated classification in terms of phases. Thus, we now present data as a whole, but offer all the available information to April 21st, 2020.

  • Line 124: Authors wrote that they used logistic regression but thy not provide any evidence of it. I would like to see some tables showing the results.

Thank you, we have added table 4, reporting logistic regression analyses.

  • None of the figures can be interpreted. The possible patterns that can be observed actually respond to a larger population in these areas. Authors should represent the standardized rates by age and sex, at a minimum (maybe also by the migration structure).

Thank you, we have moved the figures (maps) to appendix section to provide it as additional figures.

  • Authors should explain in the limitations that they do not control for all the confounders. They should mention the use of random effects in this case.

 Thank you, we have added this specific limitation at the end of the limitations section.

Minor comments:

  • Line 44: Reference 3 is missing. Done
  • Line 55: Authors wrote reference 6, but I cannot find reference 5 in the text. Done
  • Line 84: Authors should provide the web of the DGE. Done
  • Lines 84-85: Authors wrote that socioeconomic characteristics of patients and confirmation status were obtained from all reported cases. The authors should explain how they obtain this information. What are the sources? Thank you, we have rephrased, clarified and we now provide the source cited.

Reviewer 2 Report

This work presents a state-level correlation analysis for Mexico of how COVID-19 incidence, detection rate, and lethality relate to a variety of factors related to demographic and healthcare-access. The work is timely and worth publishing after minor revisions, primarily to address clarity and consistent use of terminology. The manuscript may benefit from a careful review of grammar. I hope the authors will find my remarks helpful.

Detailed comments:

Throughout, including title: What is phase 2? This usage of phases is clearly not the same as the World Health Organization's usage of pandemic phases, and it is not clearly defined in this manuscript. I suggest removing it from the title and abstract (since readers seem unlikely to be a priori familiar), and if it is an important concept, explaining what the phases are in a table. [I see in the Results that these are phases declared within Mexico, but it is still not clear what exactly they mean; in any case, they need to be defined where they are first referred to.]

Also in title—should probably be “confirmation” rather than “confirmations”

Line 31-32: “geriatric individuals were over-tested” This seems inappropriately judgmental. I think you mean that geriatric patients were tested disproportionately often, but it sounds like you are saying it was wrong to test them this much.

Line 62: None of dengue, chikungunya, or influenza should be capitalized.

Lines 62-65: “For COVID-19, sentinel surveillance does not provide certainty for patients’ contagion factor, or for defining patients’ unique clinical approach. If sentinel surveillance means a significant cost reduction and has scientific rigor, it poses a problem for correctly identifying individuals with the disease and further defining approach and treatment.” I do not understand what is meant by these two sentences. What is a contagion factor, and what does it mean to define patients' unique clinical approach? I also do not understand how surveillance creates the problem described in the second sentence.

Lines 81-82: Are some patients tested multiple times? How are patients counted who had both positive and negative tests? What laboratory tests were being used in Mexico? The RT-PCR tests that are being used generally have substantial false negative rates (~30%, or 70% sensitivity, although this varies over the course of the infection)--what effect would that have on the conclusions of this manuscript?

Throughout: “Incidence rate” is the number of cases in some unit of time, e.g. 100 cases per year. “Incidence proportion” (or “incidence per 100,000” or some other denominator) is the number of cases adjusted for the population size, e.g. 100 cases per 100,000 people. “Incidence rate ratio” seems inappropriate in this manuscript, unless you are dividing one incidence rate by another incidence rate—but that doesn't seem to be the case here.

Line 115: I don't think you mean a percentage of deaths divided by lab-confirmed cases. Maybe you mean deaths divided by lab-confirmed cases, expressed as a percentage?

Line 151: The sentence says “confirmed and untested” and then the numbers given are (older vs younger), suggesting that the confirmed cases were older. The text says confirmed patients were younger, however. Please clarify.

Lines 154-156: p can't be less than 0. Please fix this, using scientific notation if necessary. You are reporting 0 significant digits for these probabilities.

Figure 2: This type of figure is almost useless. If I want to know the demographics for patients in Puebla, how can I read those values from the figure? Why are curves plotted from state to state instead of points, when states are categorical values, and you cannot meaningfully interpolate between them? States should be sorted by whatever number is most important here, and not alphabetically by name. Right now I can find the names of states quickly, but it is very difficult to pick out most of the values. If the intent of this figure is simply to communicate the values for each state, please use a table; if there is a pattern to visualize, then please modify the figure to make the pattern apparent.

Throughout: “lethality rate,” “lethality percentage,” and “lethality” are used in different places, I think to mean the same thing. I would just use “lethality” with a clear definition.

Discussion: Citation 15 should be at the end of the previous sentence. Also, Héctor hyphenates his last name.

Line 186: “However, when the contingency from COVID-19 arose from an unexpected phenomenon, basal conditions no longer existed, and it became unknown if similar or discordant may affect incidence, confirmation, and lethality across Mexican states.” I think I know what this sentence is saying, but I'm not sure. What are basal conditions? Also, similar and discordant are adjectives, but they don't appear to be modifying any noun here.

Line 208: “validated” appears to be an extra word here.

Author Response

The Response, revised version (R2)

Dear reviewers, the authors want to thank you for your time, positive criticism, and insightfulness that has truly helped us improve our manuscript.

Please find the changes in the revised version highlighted in yellow.

Reviewer 2

The manuscript may benefit from a careful review of grammar. I hope the authors will find my remarks helpful.

We certainly did, thank you for your comments and suggestions. We sincerely appreciate them.

Detailed comments:

  • Throughout, including title: What is phase 2? This usage of phases is clearly not the same as the World Health Organization's usage of pandemic phases, and it is not clearly defined in this manuscript. I suggest removing it from the title and abstract (since readers seem unlikely to be a priori familiar), and if it is an important concept, explaining what the phases are in a table. [I see in the Results that these are phases declared within Mexico, but it is still not clear what exactly they mean; in any case, they need to be defined where they are first referred to.]

Dear reviewer, we did an extensive review, and we now understand the significance of your comment. Phases are generically applied and presented in articles depending on the country, disease agent, etc.

We have changed every mention regarding phases in our manuscript and now refer to the time when the imported transmission started and when community transmission began.

  • Also in title—should probably be “confirmation” rather than “confirmations”

Sure, we did and rephrased the title.

  • Line 31-32: “geriatric individuals were over-tested” This seems inappropriately judgmental. I think you mean that geriatric patients were tested disproportionately often, but it sounds like you are saying it was wrong to test them this much.

Thank you, you are right. We have changed accordingly.

  • Line 62: None of dengue, chikungunya, or influenza should be capitalized.

We have corrected, thank you.

  • Lines 62-65: “For COVID-19, sentinel surveillance does not provide certainty for patients’ contagion factor, or for defining patients’ unique clinical approach. If sentinel surveillance means a significant cost reduction and has scientific rigor, it poses a problem for correctly identifying individuals with the disease and further defining approach and treatment.” I do not understand what is meant by these two sentences. What is a contagion factor, and what does it mean to define patients' unique clinical approach? I also do not understand how surveillance creates the problem described in the second sentence.

Thank you, we rephrased and eliminated the paragraph as it originally was.

  • Lines 81-82: Are some patients tested multiple times? How are patients counted who had both positive and negative tests? What laboratory tests were being used in Mexico? The RT-PCR tests that are being used generally have substantial false negative rates (~30%, or 70% sensitivity, although this varies over the course of the infection)--what effect would that have on the conclusions of this manuscript?

Thank you for your comment. Patients with the WHO criteria were tested just once. Only health professionals on duty directly in contact with COVID-19 patients are considered for periodical tests on a voluntary basis. The government has banned serologic tests, and only public laboratories can perform COVID-19 diagnostic tests. Unfortunately, due to government regulations, we do not know the potential effect of sampling methods and techniques on our manuscript in relation to the Mexican population. Furthermore, private laboratories must send each COVID-19 test results to the Instituto Nacional de Referencia Epidemiologica (INRE), which is a public laboratory for “certification.”  Therefore, each test is only performed once; and patients are listed as confirmed or discarded.

  • Throughout: “Incidence rate” is the number of cases in some unit of time, e.g. 100 cases per year. “Incidence proportion” (or “incidence per 100,000” or some other denominator) is the number of cases adjusted for the population size, e.g. 100 cases per 100,000 people. “Incidence rate ratio” seems inappropriate in this manuscript, unless you are dividing one incidence rate by another incidence rate—but that doesn't seem to be the case here.

Dear reviewer, we have attended your comment by eliminating the “incidence rate” that was out of context. Nevertheless, we cannot remove Incidence rate ratios (IRR) from our regression models, as Poisson regressions express the statistic values on incidence rate ratios, and reviewer 1 has recommended us to precisely using Poisson regression. We carefully checked the use of related terms to avoid confusion and ad clarity.

  • Line 115: I don't think you mean a percentage of deaths divided by lab-confirmed cases. Maybe you mean deaths divided by lab-confirmed cases, expressed as a percentage?

Thank you, we have corrected it.

  • Line 151: The sentence says “confirmed and untested” and then the numbers given are (older vs younger), suggesting that the confirmed cases were older. The text says confirmed patients were younger, however. Please clarify.

Thank you, we have corrected and clarified in the manuscript. The original paragraph was eliminated as a result of the revision.

  • Lines 154-156: p can't be less than 0. Please fix this, using scientific notation if necessary. You are reporting 0 significant digits for these probabilities.

Thank you, we have corrected it.

  • Figure 2: This type of figure is almost useless. If I want to know the demographics for patients in Puebla, how can I read those values from the figure? Why are curves plotted from state to state instead of points, when states are categorical values, and you cannot meaningfully interpolate between them? States should be sorted by whatever number is most important here, and not alphabetically by name. Right now I can find the names of states quickly, but it is very difficult to pick out most of the values. If the intent of this figure is simply to communicate the values for each state, please use a table; if there is a pattern to visualize, then please modify the figure to make the pattern apparent.

Thank you, you are right. We eliminated figure 2. Maps are now presented as an appendix, to be offered as complementary figures.

  • Throughout: “lethality rate,” “lethality percentage,” and “lethality” are used in different places, I think to mean the same thing. I would just use “lethality” with a clear definition.

We changed the text and now use “lethality.”

  • Discussion: Citation 15 should be at the end of the previous sentence. Also, Héctor hyphenates his last name.

We have corrected, thank you.

  • Line 186: “However, when the contingency from COVID-19 arose from an unexpected phenomenon, basal conditions no longer existed, and it became unknown if similar or discordant may affect incidence, confirmation, and lethality across Mexican states.” I think I know what this sentence is saying, but I'm not sure. What are basal conditions? Also, similar and discordant are adjectives, but they don't appear to be modifying any noun here.

Thank you, we rephrased and eliminated awkward sentences.

  • Line 208: “validated” appears to be an extra word here.

We have eliminated it, thank you.
